# Targeting Integrin α3 Blocks β1 Maturation, Triggers Endoplasmic Reticulum Stress, and Sensitizes Glioblastoma Cells to TRAIL-Mediated Apoptosis

**DOI:** 10.3390/cells13090753

**Published:** 2024-04-26

**Authors:** Yuki Kuranaga, Bing Yu, Satoru Osuka, Hanwen Zhang, Narra S. Devi, Sejong Bae, Erwin G. Van Meir

**Affiliations:** 1Laboratory of Molecular Neuro-Oncology, Department of Neurosurgery, Heersink School of Medicine, University of Alabama at Birmingham, Birmingham, AL 35294, USA; ykuranaga@uabmc.edu (Y.K.); sosuka@uabmc.edu (S.O.); 2O’Neal Comprehensive Cancer Center, University of Alabama at Birmingham, Birmingham, AL 35294, USA; sbae@uabmc.edu; 3Laboratory of Molecular Neuro-Oncology, Department of Neurosurgery and Hematology & Medical Oncology, School of Medicine, Emory University, Atlanta, GA 30322, USA; bing.yu@emory.edu (B.Y.); hanwen.zhang@utsouthwestern.edu (H.Z.); ndevi@emory.edu (N.S.D.); 4Division of Preventive Medicine, University of Alabama at Birmingham, Birmingham, AL 35294, USA

**Keywords:** brain tumor, glioblastoma, integrin, ER stress, death receptor 5

## Abstract

Glioblastoma (GBM) is a devastating brain cancer for which new effective therapies are urgently needed. GBM, after an initial response to current treatment regimens, develops therapeutic resistance, leading to rapid patient demise. Cancer cells exhibit an inherent elevation of endoplasmic reticulum (ER) stress due to uncontrolled growth and an unfavorable microenvironment, including hypoxia and nutrient deprivation. Cancer cells utilize the unfolded protein response (UPR) to maintain ER homeostasis, and failure of this response promotes cell death. In this study, as integrins are upregulated in cancer, we have evaluated the therapeutic potential of individually targeting all αβ1 integrin subunits using RNA interference. We found that GBM cells are uniquely susceptible to silencing of integrin α3. Knockdown of α3-induced proapoptotic markers such as PARP cleavage and caspase 3 and 8 activation. Remarkably, we discovered a non-canonical function for α3 in mediating the maturation of integrin β1. In its absence, generation of full length β1 was reduced, immature β1 accumulated, and the cells underwent elevated ER stress with upregulation of death receptor 5 (DR5) expression. Targeting α3 sensitized TRAIL-resistant GBM cancer cells to TRAIL-mediated apoptosis and led to growth inhibition. Our findings offer key new insights into integrin α3’s role in GBM survival via the regulation of ER homeostasis and its value as a therapeutic target.

## 1. Introduction

GBM is the most aggressive brain tumor in adults, and is characterized by its cellular heterogeneity and highly invasive behavior into adjacent normal brain tissue [1]. Surgery is the initial therapeutic approach for GBM patients; however, its efficacy is limited by deeply infiltrated tumor cells that escape the resection [2,3]. The current standard of care also includes concurrent radio/chemotherapies; however, these treatments do not specifically target tumor cells, have severe dose-limiting toxicities that decrease their effectiveness, and the tumor cells rapidly develop therapeutic resistance [4]. Therefore, novel targeted therapies are desperately needed for GBM [5]. Integrins are implicated in tumorigenesis in many types of cancer and are very well suited for targeted therapies due to their cell-surface abundance and elevated expression in tumors.

Integrins are heterodimeric cell-surface receptors, consisting of α3 (ligand specificity) and β (structure) subunits. A total of 18 α and 8 β subunits give rise to 24 different integrins. The 12 αβ1 integrins can be subdivided according to their distinct ligand-binding specificities determined by their α subunits, including laminin receptors (α3, α6, and α7), collagen receptors (α1, α2, α10, and α11), Arg-Gly-Asp (RGD) receptors (α5, αv, and α8), and leucine-aspartic acid-valine (LDV) receptors (α4 and α9) [6]. Integrin-mediated anchorage to extracellular matrix (ECM) is an essential requirement for many cell functions, including growth, survival, differentiation, and migration. [7,8] Cells denied such anchorage normally undergo a distinct type of apoptosis, termed anoikis [9]. Resistance to programmed cell death is a hallmark of cancer [10,11]. Furthermore, integrins and growth factor receptors converge on similar survival signaling pathways, such as activating MAPK/ERK and PI3K/AKT survival pathways [12,13]. Additionally, integrin signaling regulates both expression and activity of apoptosis regulators, conferring abnormal resistance to apoptotic stimuli.

Numerous studies have highlighted the roles of integrins in tumor initiation, proliferation, migration invasion, and angiogenesis [14] spurring their targeting in cancer. The most prominent integrin-targeted drug is cilengitide, an RGD-mimetic antagonist of integrins αvβ3 and αvβ5, that can induce apoptosis in GBM cells and surrounding endothelial cells in preclinical cancer models. However, cilengitide failed to extend patient survival in GBM patients in a randomized, phase 3 trial [15], possibly due to enhanced tumor growth and microvascular formation at lower doses of the drug [16]. The targeting of non-RGD binding integrins in GBM growth needs to be investigated and may lead to effective alternative anti-integrin therapies. Integrin α3 shows promigration/invasion functions in breast, bladder, and prostate cancers [17,18,19]. It is overexpressed in glioma stem-like cells and promotes migration and invasion [20], raising the possibility to use integrin α3 as a therapeutic target for GBM and a prognostic marker. However, whether integrin α3 sustains GBM survival is unknown. Integrins α3, α6 and α7, when partnered with β1 function as laminin receptors. Consisting of a large class of trimeric ECM proteins, laminins play critical roles in angiogenesis, cell adhesion and migration in both normal and cancer cells. For instance, laminin-8 is associated with increased invasiveness and recurrence of GBM tumors [21]. The multifaceted roles of integrin α3 in GBM suggest a unique chance for integrin α3 inhibitors to suppress multiple pro-tumor signaling pathways, such as survival, migration, and invasion [22].

Glioma cells exhibit a heightened level of background endoplasmic reticulum (ER) stress as compared to normal cells, due to the augmented protein synthesis rate of rapidly proliferating cells and both extrinsic (hypoxia and nutrient deprivation) and intrinsic stresses (genomic instability, oncogene expression, and metabolic burden), all of which leads to heightened frequency of protein misfolding. Within the ER lumen, resident chaperones and foldases ensure correct protein folding as part of the protein maturation process, thus maintaining ER proteostasis. However, if protein folding demand exceeds ER capacity, misfolded proteins accumulate, resulting in ER stress. To restore ER proteostasis, cells trigger an adaptive unfolded protein response (UPR) via the activation of three ER transmembrane protein sensors and their downstream signaling: inositol-requiring enzyme 1α (IRE1α), activating transcription factor 6α (ATF6α), and protein kinase R–like ER kinase (PERK) [23]. Under basal conditions, all three sensors (IRE1, ATF6, and PERK) remain dormant in their inactive state due to binding with the ER-resident chaperone binding immunoglobulin protein (BiP/GRP78). However, upon ER stress, BiP detaches from the ER sensors and interacts with misfolded proteins, triggering the UPR signal transduction cascade [24].

In this study, we examined the importance of α/β1 integrins for GBM cell growth and revealed that targeting of the α3 subunit triggers ER stress and proapoptotic signaling that can be exploited for therapeutic gain.

## 2. Materials and Methods

### 2.1. Reagents

Expression vectors, short-interfering RNAs, antibodies, and chemicals are described in Appendix A.

### 2.2. Cell Culture

Human glioma cell lines (LN229, LN319, LN444, LN751, SF188) [25,26,27], human normal lung bronchial epithelial BEAS-2B cells (CRL-9609, ATCC, Manassas, VA, USA), and human embryonic kidney 293T cells were cultured on adhesive cell culture dishes in DMEM medium (10-013-CV, CORNING, Corning, NY, USA) supplemented with 10% fetal bovine serum (FBS; 10437-028, Thermo Fisher Scientific, Waltham, MA, USA) and 1% Penicillin and Streptomycin (SV30010, Cytiva, Marlborough, MA, USA) under an atmosphere of 95% air and 5% CO_2_ at 37 °C. Colorectal cancer cell line HCT116 was maintained in McCoy’s 5A Medium with 10% FBS. All cell lines were authenticated via short tandem repeat profiling and were regularly tested for mycoplasma.

For laminin experiments, cell culture-treated plates (TPN1006, Alkali Scientific, Fort Lauderdale, FL, USA) were coated with 10 or 50 µg/mL of laminin (L2020, Sigma-Aldrich, Burlington, MA, USA) according to the manufacturer’s protocol.

For non-adhesive cell culture conditions, non-cell culture-treated plates (351143, Corning) were coated with hydrogel (Poly-HEMA; poly 2-hydroxyethyl methacrylate, P3932, Sigma-Aldrich). Poly-HEMA (20 mg/mL dissolved in 95% ethanol on a rotating wheel at 42 °C for 24 h) was added to the plates and dried overnight. The number of viable cells was determined by performing the trypan blue dye exclusion test.

### 2.3. Transfection of Short-Interfering RNAs and Overexpression Plasmids

Cells were seeded in 6-well plates (0.3 × 10^6^/well) on the day before the transfection. All siRNA transfections were performed using standard Lipofectamine RNAiMAX (13778075, Thermo Fisher Scientific) transfection protocol, with 30 nM of siRNA oligonucleotides. All analyses and further treatments were performed 48 h post-transfection, unless otherwise specified. For plasmid transfection, Lipofectamine 3000 (L3000008, Thermo Fisher Scientific) was used according to the manufacturer’s protocol.

### 2.4. Cell Viability and Death Assays

The trypan blue dye exclusion test was used to quantify viable cells in 6-well dishes using a hemocytometer. Cell supernatant was diluted 1:1 with trypan blue and floating cells/field determined. Cells were detached from cell culture dishes with trypsin, then a 1/100 aliquot diluted 1:1 with trypan blue prepared and live cells/field counted. Sulforhodamine B (SRB) cell viability assay was performed as described [28]. For cell apoptosis profiling, treated LN229 cells were stained with FITC Annexin V Apoptosis Detection Kit (V13245, Thermo Fisher Scientific). Results were acquired with a LSR Fortessa flow cytometer (Becton Dickinson, Franklin Lakes, NJ, USA) and analyzed with FACSDiva software version 8.0.2 (Becton Dickinson). Caspase activity assays were performed using Caspase-Glo^®^ 8 (G8200, Promega, Madison, WI, USA) or 3/7 assay kits (G8090, Promega). Cells were first transfected with siRNA to suppress target gene expression for 48 h, then reseeded at 15,000/well in 96-well plates overnight prior to caspase assay reagent incubation for 45 min at room temperature. Cell sample luminescence was measured by a plate-reading luminometer.

### 2.5. Protein Extraction and Western Blotting

The cells were rinsed with ice-cold PBS three times and lysed in radio-immunoprecipitation assay (RIPA) buffer (89901, Thermo Fisher Scientific) supplemented with protease and phosphatase inhibitors cocktail (78442, Thermo Fisher Scientific). The cell lysate was collected with a cell scraper, sonicated with a sonic dismembrator (FB120, Thermo Fisher Scientific) at 20% amplitude for 10 s and cooled on ice immediately. Protein concentration was measured using BCA assay (23225, Thermo Fisher Scientific), and the protein lysates were diluted to 1–2 μg/μL concentration by mixing with 4x Laemmli buffer (500 mM Tris-HCl (pH 6.8), 44.4% (*v*/*v*) glycerol, 4.4% sodium dodecyl sulfate, 0.02% bromophenol blue, and 10% 2-mercaptoethanol) and PBS. Then the samples were heated at 95 °C for 5 min, cooled down and loaded on a 10% SDS-PAGE gel, and electrophoresed at 100 V/constant for 1 h. Resolved proteins were transferred to PVDF membranes (1620177, BioRad, Hercules, CA, USA) at 100 V/constant for 1 h 30 min. PVDF blots were incubated in blocking buffer (TBS; 20 mM Tris-HCl (pH 7.4), 150 mM NaCl with 0.1% Tween 20; TBS-T containing 5% nonfat dry milk (Barcode 041415012257, Publix, Lakeland, FL, USA)) at room temperature for 1 h. After three 5 min washes with TBS-T, the blots were incubated overnight with properly diluted antibodies in TBS-T containing 2% bovine serum albumin and 0.01% sodium azide at 4 °C. Subsequently, the blots were washed 3 times for 5 min in TBS-T and incubated with horseradish peroxidase-conjugated secondary antibody in the blocking buffer at room temperature for 1 h. After three more washes with TBS-T, immunoreactive bands were visualized using enhanced chemiluminescence substrate (34580, Thermo Fisher Scientific) and the blots scanned using G:BOX Chemi-XX6 and analyzed using GeneSys software version 1.8.4.0 (Synoptics Group, Frederick, MD, USA). The phospho-Akt/total Akt protein expression ratio was determined via densitometry analysis with imageJ software version 1.53k (https://imagej.nih.gov/ij/). All immunoblotting was repeated at least three times.

### 2.6. Detection of Apoptotic Nuclei

Hoechst 33342 (H3570, Thermo Fisher Scientific) was added to the cell culture medium at 10 μg/μL, then after 10 min at 37 °C, the nuclei were examined with a BZ-X810 fluorescence microscope (Keyence, Osaka, Japan).

### 2.7. Reverse Transcriptase PCR (RT-PCR)

First-strand cDNA synthesis was performed with ProtoScript II First Strand cDNA Synthesis Kit (E6560, New England BioLabs, Ipswich, MA, USA) according to the manufacturer’s protocol from 1 μg total RNA isolated by TRIzol reagent (15596018, Thermo Fisher Scientific). cDNA fragments were amplified by PCR using the primers shown in the Appendix A. PCR amplification was carried out using the Quick-Load Taq 2X Master Mix (M0271L, New England BioLabs) with an initial denaturing step at 95 °C for 5 min, then 30 cycles with 30 s denaturation at 95 °C, 60 s annealing (DR5: 53 °C, ITGA3: 55 °C, and GAPDH: 60 °C), and 60 s extension at 72 °C, ending with 5 min at 72 °C, then cooled and kept at 4 °C. The PCR products were loaded on 1% agarose gels and electrophoresed in 1x TAE buffer at 5 V/cm.

### 2.8. mRNA Stability Assay

LN229 cells were treated with actinomycin D (2 μg/mL in media, Sigma-Aldrich) 48 h post siRNA transfection. Cells were harvested at 0, 4, and 8 h after drug treatment and RNA extracted using TRIzol reagent (15596018, Thermo Fisher Scientific). Total RNA was used to perform reverse transcription PCR. Image quantification was done with ImageJ software version 1.53k. Relative DR5 mRNA levels were normalized with internal control GAPDH.

### 2.9. RNA-Seq

LN229 cells were transiently transfected with siRNAs for 48 h and total RNA was extracted using TRIzol. RNA-seq was performed at the Genomics core facility (NIH, Bethesda, MD, USA).

### 2.10. Immunoprecipitation

LN229 cells were transiently transfected with siRNAs for 48 h, then washed with cold PBS and lysed in culture dishes with IP buffer (50 mM Tris-HCl (pH 7.5), 150 mM NaCl, 1% Triton X-100, 0.1% Na-deoxycholate, and 1 mM EDTA) supplemented with protease and phosphatase inhibitors for 15 min on ice [29]. Cell lysates were scraped off dishes and briefly sonicated and their protein concentration measured via BCA assay (23225, Thermo Fisher Scientific). An amount of 500 μg of whole-cell extract proteins were incubated with antibodies against DR5 or integrin β1 (1 μg/mL) overnight at 4 °C, then 30 μL prewashed Protein G agarose slurry (sc-2002, Santa Cruz Biotechnology, Dallas, TX, USA) was added, and the mix was incubated for 2 h at 4 °C under mild rocking. Bead-bound protein complexes were washed three times with IP buffer and eluted with 2× Laemmli sample buffer by boiling for 5 min. Immunoprecipitated proteins were separated by SDS-PAGE and detected via immunoblotting.

### 2.11. Statistical Considerations

nQuery 8.4 software was used to calculate the sample size and power. Graphpad Prism 6.0 software was used to assess statistical significance. For two group comparisons, we used two-tailed Student’s t-test; with *n* = 3 replicates per group we could detect an effect size of 3.1. For comparison across groups, we used one-way analysis of variance (ANOVA) followed by Dunnett’s post hoc test and could detect an overall effect size of 1.45. With *n* = 3 in each time point in each group, for one-way repeated measures ANOVA, we could detect a difference of 1.77 in a design with 2 repeated measurements assuming a compound symmetry covariance structure, the standard deviation of 1.0, and the correlation between observations on the same sample is 0.2. For Kaplan–Meier survival analyses of the TCGA dataset, we used the log-rank test. *p* values lower than 0.05 were considered statistically significant. All histogram data represent mean ± SEM (standard error of the mean).

### 2.12. Bioinformatics Analysis

Expression analysis of integrin subunits was performed using R statistical Software (v4.2.; R Core Team 2022) and the heatmap was processed and generated using the reshape and ggplot2 R packages [30,31]. The mRNA dataset from TCGA-GBM project from 2016-01-28 release date (containing normal brain (*n* = 10) and primary IDHwt GBM (2021 WHO CNS grade 4; *n* = 387)) was obtained from Broad GDAC Firehose (https://gdac.broadinstitute.org/).

For expression and survival analyses, we used GlioVis (http://gliovis.bioinfo.cnio.es) [32] on the TCGA datasets containing primary IDHwt GBM (*n* = 357) [33]. 

For the expression profile of ER unfolded protein response-related genes, we performed RNA-seq analysis as described [34]. We selected ER unfolded protein response-related genes using the DAVID Functional Annotation Clustering Tool [35]. The heatmaps on the RNAseq data were generated using the Spotfire software package (TIBCO Software, Palo Alto, CA, USA).

### 2.13. Illustration

For Schematic summary illustration, we created the figure with BioRender.com (Toronto, ON, Canada).

## 3. Results

### 3.1. Integrin α3β1 Is Overexpressed in Cancer and Specific Silencing of Integrin α3 Induces Apoptosis

To examine the role of α/β1 integrins in GBM, we queried the Cancer Genome Atlas (TCGA) database [33] for gene expression profiles, and consistently with the prior literature [14], found that most α/β1 integrin subunits mRNAs are overexpressed in GBM as compared to normal brain (Figure 1A). To experimentally determine whether any of the 12 α/β1 integrin(s) are needed for GBM survival, we depleted each individual α subunit via siRNA-mediated knockdown in human GBM cells. Remarkably, knockdown of integrin α3-induced strong cleavage of poly-ADP ribose polymerase (PARP), a late-apoptosis marker while knockdown of other integrins had little (α2, α4, α6, and α7) or no effect (Figure 1B). Siα3 treatment caused a decrease in live cells, an increase in floating cells, and annexin V/PI flow cytometry confirmed apoptosis induction with concurrent PARP cleavage (Figure 1C). Apoptosis induction could be rescued via stable transfection of an siα3-resistant integrin α3 expression construct (see Appendix A), excluding off-target effects (Figure 1D). This cell death phenotype was observed in cancer cell lines derived from different organs (brain, breast, lung, colorectal) and with various genetic background (Appendix A). Notably, it was independent of p53 status (Appendix A, HCT116). As tumors are nutrient deprived, we further tested whether siα3 sensitized tumor cells to growth factor withdrawal and observed that serum starvation exacerbated apoptosis in α3-depleted cells (Appendix A). The siα3-induced apoptosis pathway was dependent upon cytochrome c release from the mitochondria as it was abrogated upon Bax silencing (Appendix A). 

To gain a preliminary understanding as to whether α3β1 integrin may relate to malignancy and be of value as a therapeutic target, we examined GBM patient overall survival in relation to integrin α3 (ITGA3) gene expression in the TCGA database [33]. Elevated expression of *ITGA3* was significantly associated with decreased overall survival in GBM patients (Figure 1E). We next looked at non-tumoral cells and found that siα3 had no effect on cell viability, caspase activity, and PARP cleavage in human fibroblasts (HFF-1) (Appendix A) and lung epithelial cells (BEAS-2B) (Appendix A).

In sum, these results show that siα3 selectively induces the killing of tumor cells via the rough activation of apoptosis in GBM and other cancer cell lines in a Bax-dependent and p53-independent way.

### 3.2. Apoptosis Induced by Integrin α3 Silencing Is Not Dependent on Activation of Other Integrins and Is Independent of Integrin-Mediated Survival Signaling

We next investigated the mechanism responsible for the prodeath effects of siα3. We noticed that silencing individual α integrin subunits resulted in expression changes in other subunits (Figure 1B), revealing a complex dynamic regulatory system. In particular, siα3 elicited significant upregulation of α5 and αv; however, this was not the cause of cell death, as siα5 and siαv failed to rescue cells from siα3 targeted cell death (Figure 2A,B). Interestingly, knockdown of β1 also affected α subunits, it notably increased α3 expression (Figure 2C,D), suggesting a positive feedback loop, and decreased α4 and α5 (Figure 1B, last lane), possibly suggesting heterodimerization with β1 stabilizes them.

Integrins are known to provide cell-survival signals via the engagement of the extracellular matrix; the disruption of such interactions can lead to anoikis, a form of programmed cell death triggered by caspase 8 [9]. To determine whether the proapoptosis effect of targeting α3 is due to blockage of integrin-mediated survival signaling, we treated GBM cells with laminin, the ECM ligand recognized by α3β1. As expected, β1 integrin signaling was activated in a laminin dose-dependent manner in control cells as detected by AKT phosphorylation at S473 (Figure 2C). We then reasoned that if inhibition of integrin α3β1 signaling drives siα3-induced death, knockdown of either α3 or β1 subunit would disrupt anchorage-dependence and induce anoikis. Unexpectedly, constitutive and laminin-mediated activation of Akt were mainly suppressed by siα3; as only siα3 (but not siβ1) elicited robust apoptosis, as detected by cleavage of PARP (Figure 2C). This was an intriguing finding as β1 is the only identified binding partner for integrin α3. To examine whether we had unveiled a unique α3-dependent mechanism that does not require β1, we also cosilenced α3 and β1. Surprisingly, cosilencing of β1 suppressed the proapoptotic effect of siα3 (Figure 2C), showing that integrin β1 is necessary for siα3-mediated apoptosis. These results led us to hypothesize that the proapoptotic effect triggered by siα3 involves a mechanism that is β1-dependent but does not involve disruption of canonical β1 integrin survival signaling. Instead, β1 appears to serve as a necessary downstream mediator of α3 targeting pro-apoptotic signaling.

To further verify that siα3-mediated cell death induction was indeed independent of α3β1 integrin signaling, we depleted Talin and FAK, two mediators of β1 integrin survival signaling [36]. Silencing of Talin or FAK had minimal effect on c-PARP induction and failed to block c-PARP induction by siα3 (Figure 2D).

Glioma cells are resistant to anoikis, a type of apoptosis triggered by loss of ECM contact [36,37,38]. We recreated such condition in vitro by culturing GBM cells on hydrogel coated dishes, where all types of cell adhesion are inhibited. As expected, glioma cells were resistant to anoikis under control siRNA conditions but remained susceptible to siα3-induced PARP cleavage (Figure 2E).

Overall, these results unveil α3 as a survival factor and novel target in GBM and suggest that siα3-induced apoptosis is not due to inhibition of canonical β1 survival signaling, but rather involves a distinct mechanism.

### 3.3. Silencing of Integrin α3 Increases Death Receptor 5 Expression and Extrinsic Caspase 8-Initiated Apoptosis Cascade

Since integrins are known to interact with survival/death signaling receptors on the cell surface, we next investigated whether α3 depletion could sensitize cancer cells to extrinsic apoptosis. Death receptor (DR) ligand TRAIL has attractive properties as an anti-cancer therapeutic due to overexpression of its receptors DR4 and DR5 in cancer cells [39,40,41,42]. However, many tumor cells are intrinsically resistant to TRAIL [43]; hence, new strategies to resensitize tumor cells to TRAIL are of particular therapeutic interest. We therefore examined whether integrin α3 depletion could sensitize GBM cells to TRAIL-induced extrinsic apoptosis using LN229 cells, which are intrinsically resistant to TRAIL. SF188 glioma cells (sensitive to TRAIL) and HFF-1 fibroblasts (resistant) were used as controls (Appendix A). Strikingly, siα3 led to robust resensitization of LN229 cells to TRAIL as indicated by a dramatic increase in apoptotic cells (Figure 3A and Appendix A), and cleavage of initiator caspase 8 and PARP, and downstream mediator caspase 9, whereas β1 knockdown had no impact on the cells’ inherent TRAIL resistance (Figure 3B and Appendix A).

We next investigated the mechanism underlying siα3-mediated sensitization to TRAIL. We found significant upregulation of DR5 in LN229 GBM and HCT116 colorectal cancer cells upon sia3 treatment (Figure 3C and Appendix A). TRAIL-induced cell death was DR5 dependent, as siDR5 neutralized TRAIL-induced PARP cleavage (Figure 3C, lane 7). Interestingly, siDR5 also blocked siα3-induced PARP cleavage in the absence of TRAIL (Figure 3C, lane 4), suggesting that DR5 is required for killing even in the absence of its ligand. High levels of DR5 can induce cell death without ligand binding via spontaneous assembly of the death-inducing signaling complex (DISC) and activation of the downstream apoptosis pathway [44]. Indeed, using coimmunoprecipitation (co-IP) experiments, we confirmed that siα3-mediated upregulation of DR5 led to Fas-associated protein with death domain (FADD) and caspase 8 recruitment (Figure 3D) and activation (Appendix A) in the absence of extrinsic death factors. Pharmacological inhibition of caspases 8 and 3 rescued cancer cells from siα3-mediated killing, as shown by a reduction in their cleavage and activity (Figure 3E,F and Appendix A), as well as downstream PARP processing (Figure 3E and Appendix A), and cell viability increase (Figure 3G). Having established DR5 as a critical mediator of siα3 targeted death, we further investigated the mechanism of DR5 upregulation. We found significant *DR5* mRNA upregulation in α3-depleted cells (Figure 3H), due to increased mRNA stability (Figure 3I). In contrast, DR5 protein stability was unaffected by α3 depletion (Appendix A) Thus, we evidenced a novel integrin α3-dependent mechanism of DR5 regulation.

In sum, these data establish that siα3 primes tumor cells for apoptosis via DR5 and caspase 8 activation and thereby resensitizes cells to TRAIL.

### 3.4. SiRNA-Mediated Targeting of Integrin α3 Triggers an ER Stress Response Due to Interference with Integrin β1 Maturation

Because upregulation of DR5 and induction of apoptosis in a death ligand-independent fashion can result from prolonged and unmitigated ER stress [45], we examined whether targeting integrin α3 might trigger an ER stress response. Siα3 treatment of GBM cells induced potent expression of ER unfolded protein response-related genes including *HSPA5* that codes for the ER resident chaperone binding immunoglobulin protein (BiP/GRP78) (Figure 4A). This effect was specific to α3, as silencing of the other laminin-binding integrin subunits (α6, α7) had little effect on UPR gene expression. Consistently, depletion of α3 integrin augmented BiP/GRP78 protein levels (Figure 4B, left). Use of a second non-overlapping siα3 yielded similar results, suggesting it was not an off-target siRNA artifact (Figure 4B, right). In contrast, siβ1 failed to induce BiP and DR5, again confirming the differential effects between siα3 and siβ1 inhibition and showing they extend to ER stress response induction (Figure 4B, right). Next, we considered how silencing of α3 might induce ER stress. Remarkably, we had observed that α3 depletion in glioma cells is accompanied by a prominent decrease in fully mature β1 (Figure 1B, Figure 2B and Figure 4B) and at times also an increase in the lower molecular weight form of β1 that is present just below on immunoblots (Figure 2E, Figure 3E and Figure 4B). Similar results were observed in colorectal cancer cells (Appendix A). Both bands are specific to β1 as they are strongly reduced in siβ1 treated cells (Figure 1B) and the lower form likely represents a pool of newly synthesized β1 undergoing post-translational modification. β1 maturation involves protein glycosylation [46] in the ER and Golgi apparatus and disruption of this process with an N-glycosylation inhibitor (kifunensine) yields lower molecular weight β1 isoforms that are similar in size to the ones seen in siα3-treated cells (Appendix A) [47]. Incomplete protein maturation leads to protein misfolding, a source of ER stress [48]. Therefore, we hypothesized that siα3 interferes with β1 maturation, preventing its proper folding, and thereby triggering ER stress. This model explains why siβ1 does not induce apoptosis and why it neutralizes siα3 proapoptotic effects (Figure 2C). Activation of the UPR was unique to tumor cells as there was no BiP increase in siα3-treated BEAS-2B cells (Figure 4B). BiP is bound to PERK1, ATF6, and IRE1 in unstressed cells, and to three ER stress receptors present on the ER membrane [49]. When misfolded proteins accumulate, they compete for BiP binding, resulting in BiP detachment from the ER stress sensors. Once BiP is unbound, PERK1, ATF6, and IRE1 trigger ER stress response signaling and overexpression of BiP can neutralize it [24,50]. To test for the involvement of PERK1 and IRE1 in the ER stress response triggered by α3, we treated GBM cells with PERK1 and IRE1 inhibitors. Pharmacological inhibition of IRE1 signaling prevented the accumulation of BiP and DR5 expression, while inhibition of PERK1 had more modest effects (Figure 4C).

The above findings suggest that β1 is a downstream mediator of siα3 proapoptotic effects and that the mechanism involves IRE1-mediated ER stress triggered by misfolded β1, and this is specific to cancer cells.

Altogether, our manuscript results show that targeting integrin α3, but not other integrin α units, blocks the maturation of integrin β1. Accumulation of immature β1 triggers unmitigated ER stress and subsequent cancer cell death in a DR5-dependent fashion.

## 4. Discussion

Due to their abundance, disease specificity, and cell-surface presence, integrins are highly suitable as therapy targets. Integrins have already been successfully targeted for the treatment of thrombosis and autoimmune diseases using ligand-mimetic antagonists or using ligand-blocking antibodies against α4 or αM [6]. Antibodies targeting α4 (natalizumab) or α4β7 (vedolizumab) have been used to maintain clinical remission in patients with inflammatory bowel disease (IRB) and a natalizumab biosimilar has been investigated in clinical trials for patients with relapsing–remitting multiple sclerosis (NCT04115488) [51]. Currently, anti-integrin therapies are increasingly being investigated in clinical trials for a variety of tumors and other diseases [52]. However, no successful integrin-targeted therapy is available for GBM, mainly due to our limited knowledge of GBM integrin biology. Previous focus on targeting angiogenesis-related integrins (αvβ3 and αvβ5) with the cilengitide peptide failed to produce survival benefits in GBM patients (NCT00689221, terminated in Phase 3), possibly linked to unforeseen complications of increased angiogenesis at lower doses of inhibitor [15]. A phase 1 clinical trial (NCT04608812) with an antibody targeting β1 (OS2966, [53]) was also terminated due to slow enrollment and financial constraints. Here we provide proof-of-principle evidence that integrin α3 has potential as a new integrin target in GBM and possibly other tumors.

Cancers, including GBM, exhibit elevated levels of ER stress signaling to cope with uncontrolled growth and an unfavorable microenvironment. Standard radio/chemotherapies have been shown to induce ER stress; this feature represents an Achille’s heel that can be exploited with new sensitizers to tip cancer cell towards cell death. A prior study showed that ritonavir (RTV), a retroviral protease inhibitor-induced ER stress, reduced migration and increased overall survival in combination with temozolomide in a GBM xenograft mouse model [54]. Our present study demonstrates that blocking α3 expression induces ER stress, specifically via the accumulation of misfolded integrin β1. Traditionally, the goal of integrin targeting has been to interrupt ligand-dependent survival signals. We unveil an important non-ligand-mediated new pro-survival role for integrin α3 that does not depend on classical α3β1 integrin signaling.

Our data support a novel non-canonical α3 integrin-mediated function, i.e., we have demonstrated an essential role of α3 in mediating β1 maturation in the ER and have shown that, in its absence, the accumulation of immature β1 triggers heightened ER stress and apoptosis via upregulation of DR5 and activation of caspase 8 (Figure 5). This was a surprising finding, as the integrin β1 structural subunit can dimerize with multiple α subunits, yet we found that this function was unique to α3 (silencing other α-units had little effect), suggesting a unique vulnerability in integrin trafficking.

Thus, our study has unveiled a novel therapeutic opportunity by disrupting integrin α3-mediated ER homeostasis, which primes tumor cells for apoptosis via the death receptor extrinsic signaling pathway and resensitizes them to TRAIL-mediated cell death.

Additionally, our research has provided evidence of the reprogramming of global αβ1 integrin expression after genetic suppression of single subunits, warranting further studies to unravel interconnectedness of α and β1 gene regulation and/or protein stability. These findings also suggest that caution will need to be exercised with new therapies targeting integrins via genetic silencing [55] rather than antagonistic inhibition; this is due to unintended changes in global integrin expression, which will result in altered cancer cell responses within the tumor microenvironment.

Another important finding of our research is the suggestion that integrin α3 is required for the maturation of integrin β1, i.e., it has a unique role among its 12 different α subunits. This role is likely due to the comaturation of integrin heterodimers. This dimerization process occurs from the early stages of integrin maturation via the ER–Golgi secretory pathway [56]; the lack of such an interaction might disrupt protein maturation, affecting proper integrin conformation and function.

Currently, no integrin α3 inhibitors are available, due to a lack of simple ligand-binding motifs for antagonists. However, recent advances in siRNA–spherical nucleic acid (SNA) nanoparticle conjugates have successfully targeted GBM via systemic administration in animal models [55,57]. The siRNA conjugates were shown to penetrate the blood–brain barrier (BBB) and to disseminate throughout orthotopic glioma implants with no side effects. This study provides the impetus for the future testing of siα3 SNAs in orthotopic glioma models in conjunction with TRAIL treatment with the objective of further translating them to novel therapeutics in human patients.

## Figures and Tables

**Figure 1 cells-13-00753-f001:**
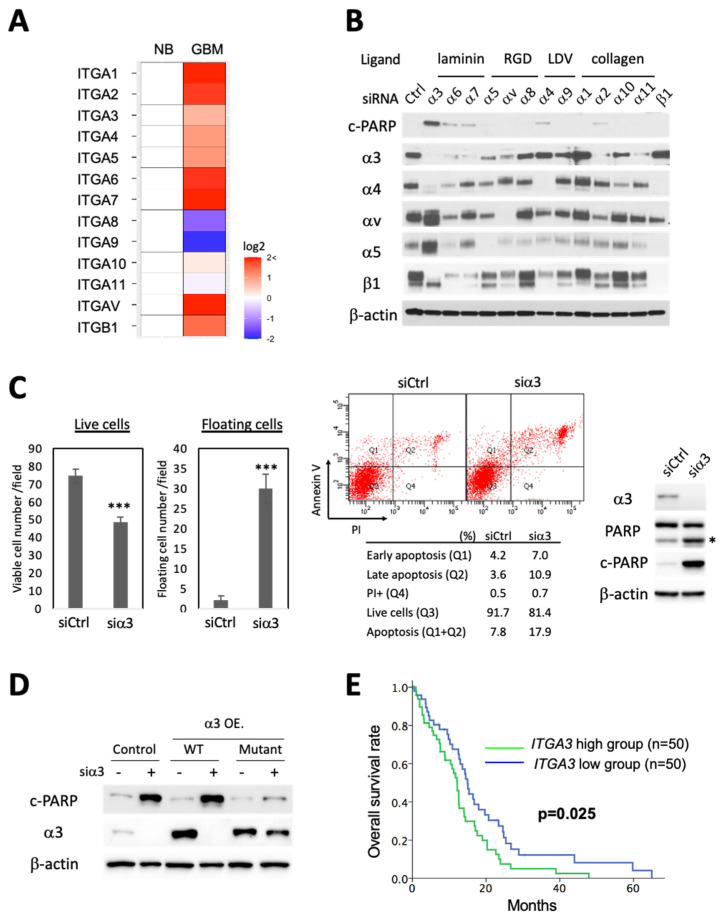
Integrin α3 promotes glioma survival. (**A**) α/β1 integrin subunit mRNA expression profile in GBM (2021 WHO CNS grade 4; IDHwt; *n* = 387) vs. normal brain (NB; *n* = 10) from TCGA dataset (2016-01-28 release date); (**B**) Immunoblot analysis of LN229 GBM cells 72 h after transfection with indicated integrin subunit siRNAs (30 nM). Cells were switched to a serum-free medium 24 h before cell lysis. Integrin α subunits are grouped by ligand specificity. (**C**) Cell viability/death analyses of LN229 cells transfected with siCtrl or siα3 (30 nM) for 96 h. Cell culture supernatant was collected, and floating cells were stained using trypan blue. Stained dead cells were counted in a hemocytometer (left panel, floating cells). Adherent cells were detached by trypsin and their viability evaluated via trypan blue staining. Average number of unstained viable cells counted per square in the hemocytometer are shown (left panel, live cells), student’s *t*-test (***; *p* < 0.001). Annexin V/propidium iodide (PI) flow cytometry analysis (middle panel). Immunoblot analysis of the flow samples (right panel). *; cleaved form of PARP; (**D**) Immunoblot analysis of LN229 cells +/− stable expression of wild type or siα3-resistant (mutant) α3 integrin treated with siα3 or siCtrl for 96 h. Note that rescue with siα3-resistant (mutant) α3 integrin prevents induction of PARP cleavage; (**E**) Kaplan–Meier survival curves of GBM (2021 WHO CNS grade 4; IDHwt) patients with lowest or highest (50 cases each) *ITGA3* mRNA expression using TCGA dataset (*n* = 357).

**Figure 2 cells-13-00753-f002:**
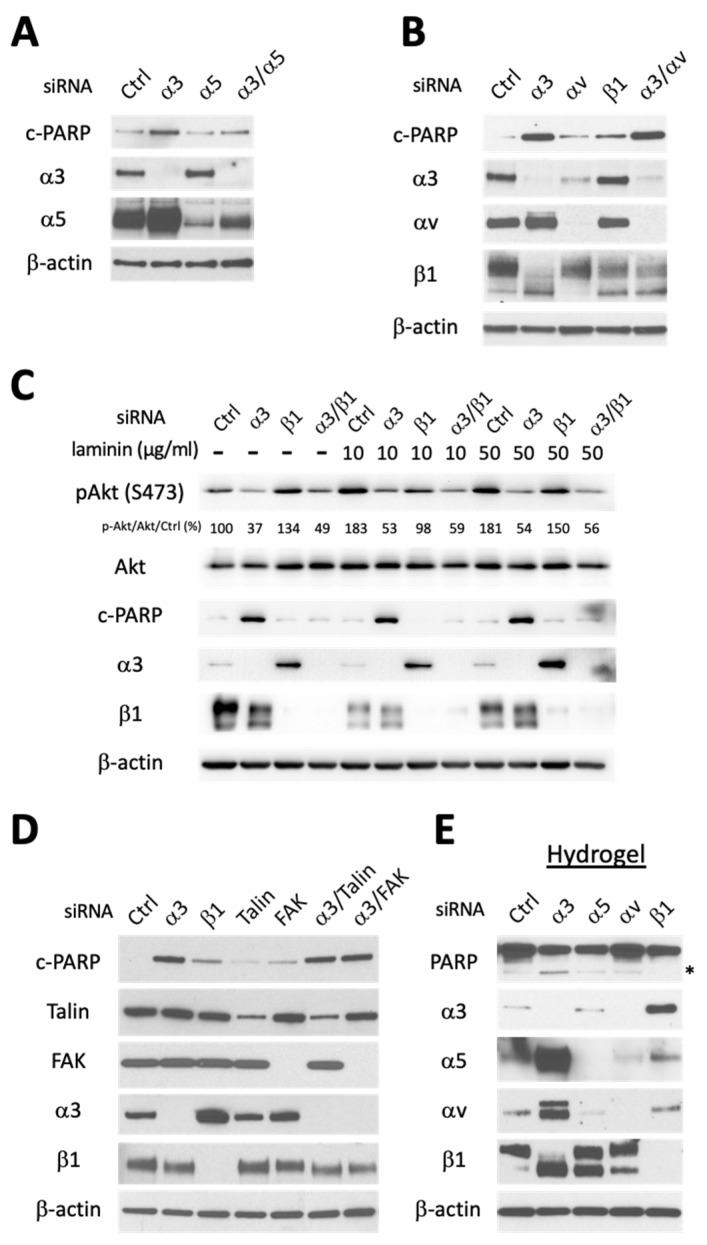
Integrin α3-targeted killing is independent of α5, αv, β1 integrin signaling, and anoikis induction. LN229 human GBM cells were transfected with indicated siRNAs (30nM) and cell extracts analyzed by immunoblotting 48 h or 72 h later. PARP cleavage was used as apoptotic marker and Phospho-Akt (S473) as activation marker for β1 integrin signaling. (**A**) siRNA-mediated integrin α5 codepletion does not rescue siα3-induced c-PARP. Cells were switched to serum-free medium 48 h after transfection and harvested 24 h later; (**B**) siRNA-mediated cotargeting of integrin αv fails to rescue siα3-induced apoptosis. Cells were harvested 72 h after transfection; (**C**) siRNA knockdown of β1 integrin rescues siα3-induced apoptosis. Cells were cultured on dishes +/− laminin (α3 ligand) coating (10 or 50 μg/mL). Cells were switched to serum-free medium 24 h after siRNA transfection and harvested 24 h later. The number underneath of the pAkt panel represents the quantification of the ratio of pAkt/total Akt/siCtrl; (**D**) siRNA-mediated knockdown of FAK or Talin (integrin α3/β1 or β1 signaling core components) does not phenocopy targeting integrin α3 subunit alone. Cells were treated with TRAIL (100 ng/mL) at 24 h post-siRNA transfection; (**E**) α3 knockdown activates PARP cleavage in cells grown under anchorage-independent conditions. Cells were transfected with indicated siRNAs for 24 h, then transferred to non-adhesive Poly-HEMA hydrogel coated cell culture dishes for 36 h. *; cleaved form of PARP.

**Figure 3 cells-13-00753-f003:**
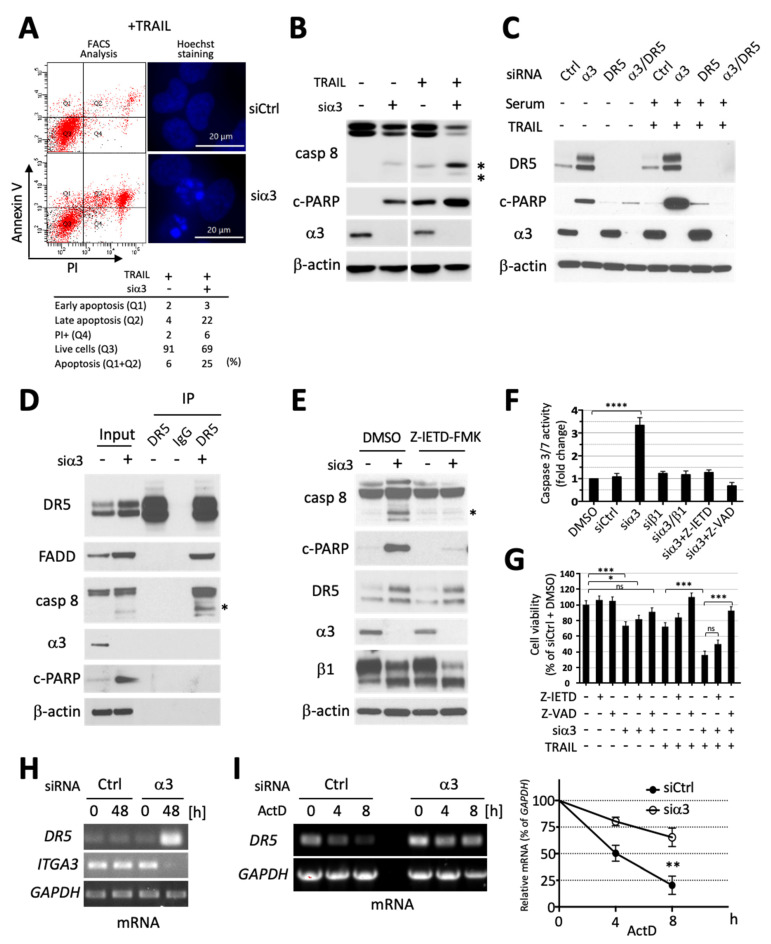
DR5 mediates integrin α3-targeted cell killing. (**A**) (Top left) Annexin V/PI flow cytometry analysis and (top right) Hoechst 33342 staining of LN229 transfected with siCtrl or siα3 (30 nM; 72 h) with TRAIL treatment (100 ng/mL, added 24 h after siRNAs transfection). (Bottom) Quantification of the FACS analysis; (**B**) Immunoblot analysis of LN229 cells transfected with siα3 or siCtrl (30 nM) for 72 h. TRAIL (100 ng/mL) was added 24 h post-transfection. Note that siα3 sensitizes cells to TRAIL-mediated cleavage of Caspase 8 (* cleaved form) and PARP; (**C**) Immunoblot showing that DR5 knockdown rescues siα3-induced LN229 cell death regardless of TRAIL treatment. TRAIL (100 ng/mL) was added 48 h post-siRNA (30nM) transfection and cells harvested 24 h later; (**D**) Co-immunoprecipitation analysis with anti-DR5 antibody suggests that siα3 (30 nM; 48 h) transfection induces DISC assembly in LN229 cells as evidenced by FADD and caspase 8 co-precipitation. Rabbit IgGs were used as controls; (**E**) Immunoblot showing that caspase 8 inhibitor (Z-IETD-FMK; 20 µM; added 4 h after siRNA transfection) rescues siα3 (30 nM; 48 h)-induced PARP and Caspase 8 cleavage in LN229 cells; (**F**) Caspase-Glo^®^ 3/7 assay shows siα3-mediated activation of caspase 3/7 in LN229 cells 48 h after siRNA transfection. Inhibitors of caspase 8 (Z-IETD-FMK, 20 μM) and pan-caspase (Z-VAD-FMK, 20 µM; 4 h post-transfection) neutralized the effect. *n* = 3. One-way ANOVA (****; *p* < 0.0001); (**G**) Cell viability of LN229 cells transfected with siCtrl or siα3 (30 nM) at 96 h with or without TRAIL treatment at 24 h post-transfection. Cells were treated with caspase inhibitors 1 h before transfection. *n* = 4. One-way ANOVA Tukey’s test (*; *p* < 0.05, ***; *p* < 0.001, ns; not significant); (**H**) Reverse transcription PCR (RT-PCR) showing siα3 (30nM; 48 h) transfection induces DR5 mRNA upregulation in LN229 cells; (**I**) DR5 mRNA stability experiment. (Left panel) LN229 cells were transfected with siα3 (30 nM; 48 h), then treated with actinomycin D (2 µg/mL) for 0, 4, and 8 h and the mRNA levels analyzed using RT-PCR. (Right panel) Quantification of cDNA band intensity by ImageJ. *n* = 3. Student’s *t*-test (**; *p* < 0.01).

**Figure 4 cells-13-00753-f004:**
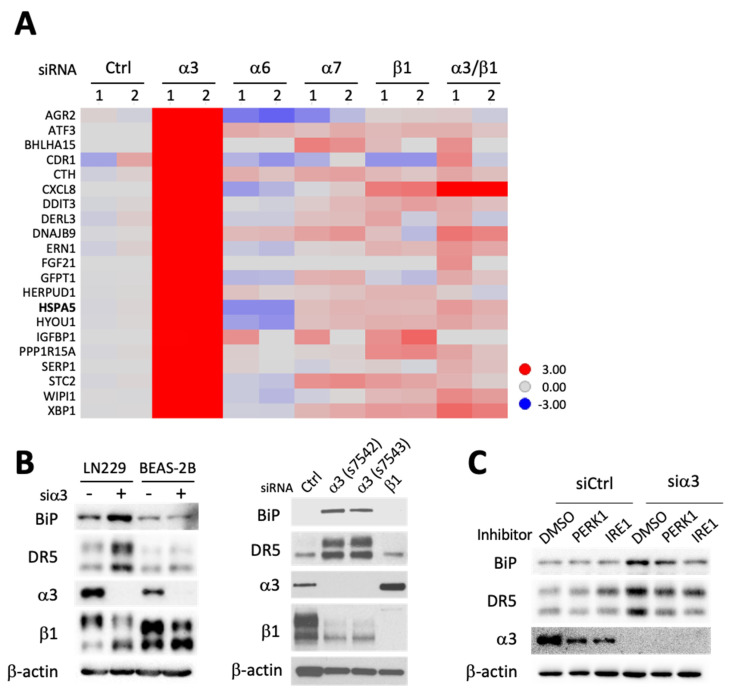
Integrin α3 suppression induces ER stress. LN229 cells were transfected with indicated siRNAs (30 nM) for 48 h and extracted mRNAs or proteins analyzed using RNAseq or immuno-blotting, respectively. (**A**) Expression profile of ER unfolded protein response-related genes in indicated siRNA-treated cells. RNAseq (*n* = 2); (**B**) (left panel) siα3 transfection induces DR5 expression in LN229 cells, but not in normal bronchial epithelial BEAS-2B cells. (Right panel) Serum-free medium was added 24 h before harvesting the LN229 cells. Two different siα3 RNAs (s7542 and s7543) were used to exclude non-specific effects. Note that both siα3 RNAs prevent β1 integrin maturation and induce ER stress marker BiP; (**C**) Pretreatment (1 h) with PERK and IRE1 signaling inhibitors (Salubrinal and STF-083010, respectively; 100 µM) prevents BiP and DR5 activation following siα3 transfection.

**Figure 5 cells-13-00753-f005:**
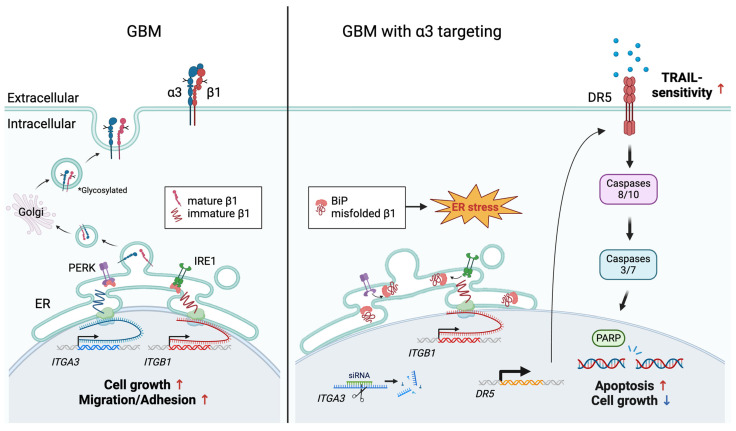
Schematic summary.

## Data Availability

The experimental data supporting the findings of this study are available within the paper and its Appendix A.

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
