# Peer review of "Targeting Integrin α3 Blocks β1 Maturation, Triggers Endoplasmic Reticulum Stress, and Sensitizes Glioblastoma Cells to TRAIL-Mediated Apoptosis"

_cells, 2024, doi:10.3390/cells13090753_

Round 1

Reviewer 1 Report

Comments and Suggestions for Authors

In the study "Targeting Integrin α3 blocks β1 maturation, triggers ER stress, and sensitizes glioblastoma cells to TRAIL-mediated apoptosis," Kuranaga et al. conducted a screening analysis of 12 α/β1 integrins for human GBM cell survival. They found that elimination of α3 induced apoptosis accompanied by strong cleavage of PARP, observed exclusively in tumor cells and not in non-tumoral cells. The authors demonstrated that siα3-mediated cell death induction was dependent on Bax and integrin β1 but independent of p53 and α3β1 integrin signaling. The study sheds light on novel, complicated interactions between α3 and β1. Moreover, the authors identified that silencing of α3 primes tumor cells for apoptosis through DR5 and caspase-8 activation, thereby resensitizing cells to TRAIL. Finally, they highlighted the importance of ER stress as a cause for cell death, presumably induced by immature, misfolded β1 integrin.

The authors employed robust experimental approaches to address several fundamental questions. I recommend that the manuscript be accepted for publication after addressing the following minor concerns.

Specific comments:

  • In my opinion, the current version of the abstract appears somewhat unbalanced. To enhance clarity and effectiveness, I recommend decreasing the background description and emphasizing the authors' findings more prominently.

  • The transition between Results 3.3 and 3.4 appears somewhat disjointed. Initially, it was unclear why the authors got interested in ER stress. Revisiting the transition sentences could enhance the coherence of the narrative. To streamline the flow, consider integrating elements from the first half of the second paragraph ('Misfolded proteins… induces ER stress[41]') and possibly 'Therefore, we hypothesized that siÉ‘3 interferes with β1 maturation, preventing its proper folding, triggering ER stress, and leading to tumor cell death.' I recommend restructuring for better continuity.

  • Page 5, Line 230: "As expected, β1 integrin signaling was activated in a laminin dose-dependent manner in control cells as detected by AKT phosphorylation at S473 (Fig. 2C)," - In Fig 2C, the trend looks not immediately evident from the figure as it stands. The authors may consider summarizing the quantification data in a separate figure panel, for example, using bar or line plots, to enhance clarity.

  • Page 5, Line 240: "These results led us to hypothesize that the pro-apoptotic effect triggered by siÉ‘3 involves a mechanism that is β1-dependent, but independent of canonical β1 integrin survival signaling, with β1 playing a role as a downstream mediator of É‘3 targeting." This sentence was not easy to understand. Restructuring should be considered for better readability.

Page 6, Line 284: "Pharmacological inhibition of caspase 8 rescued cancer cells from siɑ3 mediated killing, as shown by reduced caspase 8 cleavage (Fig. 3E) and activity (Suppl. Fig. 2E), as well as downstream PARP processing (Fig. 3E) and effector caspase 3 activity (Fig. 3F)." If the authors confirmed the reduced tumor-killing effect not only through caspase-8 cleavage and PARP processing but also through direct assessments of apoptosis, such as cell viability, presenting such data would be more helpful.

  • Page 6, Line 288: "We found significant DR5 mRNA upregulation in É‘3 depleted cells (Fig. 3G), due to increased mRNA stability (Fig. 3H). In contrast, DR5 protein stability was unaffected by É‘3 depletion (Suppl. Fig. 2F) Thus, we have shown novel integrin É‘3-dependent mechanism of DR5 regulation." The intended message in this sentence is unclear. I recommend restructuring it for clarity and conciseness.

  • Page 6, Line 312: "The process of β1 maturation involves protein glycosylation, which is essential for integrin sorting and signaling.[39,40] Glycosylation is critical for protein processing through the ER and Golgi apparatus; and its disruption leads to protein misfolding and induces ER stress[41]. " Unless the authors have confirmed the changes of glycosylation were indeed induced, the interpretation of the lower molecular weight form of β1 contains some speculations. Is there any way to confirm it was indeed due to the change in glycosylation status? If not, such speculative parts need to be clarified or may be better to be moved to the Discussion part.

  • To ensure generalizability, confirming the findings in multiple cell lines would be beneficial. If this is not feasible, it's important to provide clarification in the Discussion section, addressing the limitations of the study.

  • Instead of image analysis-based quantification, quantitative RT-PCR, such as TaqMan assay, could be considered for quantifying the RT-PCR experiment results, if feasible.
  •  
  • As an option, a cartoon summarizing the findings might be useful to facilitate readers' better understanding.
  • Consistency is needed throughout the manuscript in the placement of citation marks (such as "[1]" or "[2-3]"), whether they appear before or after punctuation marks.

  • For reproducibility, Method section descriptions need to be improved, such as RNA-seq data processing and analyses, and image data acquisition and analysis including ImageJ.

  • In Figure 1C, "sia3" should be corrected to "siÉ‘3".
Comments on the Quality of English Language

Described above. 

Author Response

Thank you very much for taking the time to review our manuscript. Please find the detailed responses in the attached file and the corresponding revisions: corrections indicated by page/line.

Reviewer 2 Report

Comments and Suggestions for Authors

Thank you for the opportunity to read this interesting work. I congratulate the authors on this great work, which I enjoyed reading. The manuscript is clearly formulated and the experiments make sense in the order in which they are presented. The outcomes of these experiments offer compelling motivation for forthcoming projects, suggesting the necessity for publication of this work. While I didn't identify any substantive errors in the content, integrating current literature through additional references could underscore the robustness of the findings and align them more closely with contemporary research paradigms. Additionally, I advocate for the development of an illustration to elucidate the newly delineated pathway described herein, enhancing comprehension and facilitating dissemination of the results.

Introduction: “Current radio/chemotherapies do not specifically target tumor cells and thus have severe dose-limiting toxicities that decrease their effectiveness“

> Another important mechanism is the occurrence of treatment resistance. This should be mentioned here.

Introduction: “Ankoisis resistance is a hallmark of cancers, including adult GBM“
> At this point, I would recommend inserting a reference, e.g. PMID: 36046036.

Results: Due to the number of experiments and the complexity of the pathways described here, the authors should provide a schematic illustration. The canonical pathway of a3/b1 and the non-canonical b1-independent pathway postulated by the authors should be outlined.

Results: “Furthermore, sia3 induced BiP/GRP78 at the protein level”. That's too colloquial.

Discussion: “Integrins have already been successfully targeted for the treatment of thrombosis and autoimmune diseases using ligand-mimetic antagonists or by ligand blocking antibodies against a4 or aM. However, no successful integrin-targeted therapy is available for GBM, mainly due to our limited knowledge of GBM integrin biology”
> Nevertheless, I would like to point out that anti-integrin therapy is also increasingly being investigated clinically in oncology for extra-cranial tumor diseases, e.g. PMID: 36588107

Discussion: “Our current study provides the impetus for the future testing of sia3 SNAs in orthotopic glioma models in conjunction with TRAIL treatment with the objective to further translate them to novel therapeutics to human patients”.
> This is a plausible next step. Given that conventional therapy for glioblastoma inherently triggers a level of ER stress, supplementing with integrin
α3 blockade could further promote cellular apoptosis. It would be beneficial at this point to juxtapose your findings with those of other studies that have explored similar potential outcomes, e.g. PMID: 33091717; 3692797. 

Author Response

(The authors gave the same response as above.)

Round 2

Reviewer 2 Report

Comments and Suggestions for Authors

Well done, from my point of view now acceptable for publication.